# STAT2 Promotes Tumor Growth in Colorectal Cancer Independent of Type I IFN Receptor Signaling

**DOI:** 10.3390/curroncol32120707

**Published:** 2025-12-16

**Authors:** Jorge Canar, Madeline Bono, Amy Alvarado, Michael Slifker, Giovanni Sitia, Ana M. Gamero

**Affiliations:** 1Department of Medical Genetics and Molecular Biochemistry, Lewis Katz School of Medicine, Temple University, Philadelphia, PA 19140, USA; jorge.canar@temple.edu (J.C.); madeline_bono@dfci.harvard.edu (M.B.);; 2Biostatistics and Bioinformatics Facility, Fox Chase Cancer Center, Philadelphia, PA 19111, USA; 3Division of Immunology, Transplantation and Infectious Diseases, IRCCS San Raffaele Scientific Institute, 20132 Milan, Italy; sitia.giovanni@hsr.it

**Keywords:** STAT2, interferon, colorectal, cancer, tumor, TCGA

## Abstract

Our study investigated the role of the protein STAT2 in colorectal cancer. STAT2 is critical in the body’s defense by activating the type I interferon (IFN) pathway, which can restrict tumor growth and viral infections. However, recent studies, including ours, find that STAT2 can promote tumor growth. In this study, we found that higher levels of STAT2 in colon cancer were linked to poorer patient survival. Removing STAT2 from tumor cells slowed tumor growth, while increasing STAT2 made tumors grow faster. We then tested whether STAT2’s tumor-promoting effects depended on the type I IFN pathway. Removing the receptor for type I IFN caused tumor cells to grow rapidly, showing that STAT2 can drive tumor growth independently of this pathway. These results suggest that STAT2 plays a previously unrecognized role in promoting colorectal cancer and could be a potential target for new cancer treatments.

## 1. Introduction

Signal Transducer and Activator of Transcription (STAT) are critical mediators of cellular responses to cytokines and growth factors, regulating proliferation, differentiation, survival, and immune functions [1]. Among these, STAT2 is best known for its exclusive and key role in the type I interferon (IFN-I) signaling pathway [2]. Following IFN-α/β binding to the IFNAR1/IFNAR2 receptor complex, receptor-associated tyrosine kinases JAK1 and TYK2 phosphorylate STAT1 and STAT2, which then dimerize and associate with IRF9 to form the IFN-stimulated gene factor 3 (ISGF3) transcriptional complex [3]. ISGF3 translocates to the nucleus and activates transcription of IFN-stimulated genes that drive immunomodulatory, antiproliferative, and antiviral responses [4,5].

Classical IFN-I signaling has long been considered tumor-suppressive [6,7]. IFN-I activation can inhibit tumor cell proliferation, induce apoptosis, and enhance antitumor immune surveillance. In particular, IFN-I signaling augments the cytotoxic activity of natural killer (NK) cells [8] and cytotoxic T lymphocytes, further contributing to tumor suppression [9,10,11]. Clinically, IFN-α has been used as an adjuvant therapy for certain cancers, including melanoma and renal cell carcinoma, reflecting its antiproliferative and immunostimulatory potential [12]. However, prolonged or dysregulated IFN signaling can also induce immune exhaustion and resistance mechanisms [13,14], revealing that its effects may be context-dependent.

The role of STAT2 in mediating the biological effects of type I IFNs is well-established [2]. However, its direct role in cancer remains underexplored. Recent studies from our group and others have reported that STAT2 may paradoxically promote tumor progression under specific conditions [15,16,17]. Elevated STAT2 expression has been associated with poor prognosis in several cancers [18,19,20], and experimental evidence indicates that STAT2 can support tumor cell survival and chemoresistance through both canonical and non-canonical signaling mechanisms [21,22,23,24]. In colorectal cancer (CRC), a malignancy where inflammation and immune modulation are key drivers of disease progression [25], the role of STAT2 remains poorly defined.

Colorectal cancer (CRC) remains a major global health concern [26] and is characterized by complex interactions between tumor cells and the tumor microenvironment [27]. Given the central importance of immune and cytokine networks in CRC progression, elucidating the contribution of STAT2 in this setting is of significant relevance. In this study, we investigated the relationship between STAT2 expression and patient survival in colorectal cancer and examined the functional role of STAT2 in tumor growth using preclinical tumor models. We further evaluated whether STAT2’s tumor-promoting activity depends on canonical IFN-I signaling. Our findings demonstrate that cell-autonomous STAT2 enhances colorectal tumor growth and correlates with poorer outcomes through mechanisms independent of IFN-I signaling, highlighting STAT2 signaling as a potential therapeutic target in CRC.

## 2. Materials and Methods

### 2.1. Patient Survival Analysis

Publicly available datasets of TCGA-colon adenoma carcinoma (COAD) cases were obtained from the Genomic Data Commons (GDC) Data Portal [28]. Data was analyzed by conducting a Kaplan–Meier survival analysis to determine overall patient survival in tumors with low and high STAT2 mRNA levels. Tumors with high STAT2 mRNA levels were further stratified by low and high IFNAR1 mRNA levels to evaluate differences in overall survival. The TCGA-COAD datasets in UALCAN were used to obtain relative expression levels of IFNAR1 and STAT2 in normal colon and COAD [29,30].

### 2.2. Cell Lines

The human HCT116 cell line (a gift from Dr. Bert Vogelstein, Johns Hopkins University) was cultured in RPMI-1640 media supplemented with 5% heat-inactivated fetal bovine serum (FBS), 2 mM L-glutamine, 100 IU/mL penicillin-streptomycin (Gibco, Grand Island, NY, USA), 0.05 mM 2-mercaptoethanol (Gibco), 1X non-essential amino acids (Genesee, El Cajon, CA, USA), and 5 µg/mL plasmocin prophylactic (InvivoGen, San Diego, CA, USA). Murine MC38 and MC38 IFNAR1 KO cell lines [31] were cultured in DMEM media supplemented with 10% heat-inactivated FBS, 2 mM L-glutamine, 100 IU/mL penicillin-streptomycin (Gibco), and 5 µg/mL of plasmocin prophylactic (InvivoGen). All cells were cultured at 37 °C and 5% CO_2_.

### 2.3. Establishment of STAT2 and IFNAR1 Knockouts for Comparative Functional Studies

Two plasmids were generated using the pSpCas9(BB)-2A-GFP (PX458) vector, each expressing a CRISPR guide RNA (gRNA) targeting either human IFNAR1 or mouse *Stat2*. The guide RNAs were purchased from GenScript (Piscataway, NJ, USA). Human STAT2 CRISPR/Cas9 KO plasmid (cat#sc-417454) was purchased by Santa Cruz Biotechnology (Santa Cruz, CA, USA). HCT116 cell lines were transfected with either IFNAR1 gRNA or STAT2 gRNA vectors. MC38 parental cells were transfected with the mouse Stat2 gRNA vector. GFP-positive cells were cell sorted by flow cytometry and expanded in culture. Single-cell clones were established by limiting dilution and screened for loss of protein expression by western blot analysis.

### 2.4. Cell Proliferation Assay

Cell proliferation was measured using the CellTiter 96 AQ_ueous_ One Solution Proliferation Assay (Promega, Madison, WI, USA). Cells (2 × 10^3^/well) were seeded in flat-bottomed 96-well plates with 100 μL of culture medium. Every 24 h, each well was treated with 20 μL of CellTiter 96 AQ_ueous_ One Solution and incubated for 3 h at 37 °C and 5% CO_2_. The absorbances at 490 nm were measured using a VictorTMX5 multilabel plate reader (Perkin Elmer Life Sciences, Waltham, MA, USA). Background values were first subtracted from each well before proceeding with data analyses. Fold change was calculated by compounding all fold change counts from day 0 and every 24 h. Three independent experiments were performed with technical replicates in quadruplicate.

### 2.5. In Vivo Tumor Studies

Wild-type and *Rag1KO* mice on the C57BL/6J background were bred in our animal facility under a pathogen-free environment. HCT116, HCT116 IFNAR1 KO, and HCT116 STAT2 KO tumor cells were resuspended in endotoxin-free 0.9% saline solution and mixed with matrigel (cat#354230; Corning Life, Tewksbury, MA, USA) at a 3:1 ratio. Six–eight-weeks-old *Rag1KO* mice received a subcutaneous injection of 5 × 10^6^ in 200 µL volume on a shaved dorsal flank. MC38, MC38 IFNAR1 KO, and MC38 STAT2 KO tumor cells, prepared at a density of 1 × 10^6^ cells, were injected s.c. on the flanks of wild-type B6 mice at a volume of 200 µL. Tumor measurements were taken using a digital caliper every 2–3 days. Tumor volume was calculated with the following formula: V  =  (*a^2^ * × *b*), where *a* is the shorter diameter and *b* is the longer diameter of the tumor. Study was terminated when tumors reached a size of 15 mm in diameter.

### 2.6. Antibodies and Cytokines

Anti-STAT1 antibody (Cat#10144-2-AP), anti-STAT2 antibody (cat##51075-2-AP), anti-STAT3 (cat#10253-2-AP), anti-β-tubulin (cat#10094-1-AP), and anti-β-Actin (cat#66009-1-Ig) were purchased from Proteintech (Rosemont, IL, USA. Anti-phospho-STAT1 Y701 (cat#9167), anti-phospho-STAT2 Y690 (cat#88410), and anti-phospho-STAT3 Y705 (cat#9145) were purchased from Cell Signaling Technology (Danvers, MA, USA. HRP-conjugated anti-mouse IgG (cat#7076) and anti-rabbit-IgG (cat#7074) were purchased from Cell Signaling Technologies. Human IFN-α was purchased from Peprotech (cat#300-02AA; Cranbury, NJ, USA). Recombinant murine IFN-β was generously provided by Biogen-Idec.

### 2.7. Western Blot Analysis

Cells were left untreated or stimulated with IFN-α (1000 U/mL) for various time points for one hour. Cells were washed in 1x PBS and disrupted in lysis buffer as described previously [32]. Protein extracts were resolved on precast SurePAGE 4–12% gradient gels (GenScript) and transferred onto PVDF membranes. Membranes were blocked with Blocker Casein TBS (Bio-Rad, Hercules, CA, USA) and incubated overnight with the corresponding primary antibodies, followed by washing and a 30 min incubation with HRP-conjugated secondary antibodies prepared in TBS + 3% BSA. Signal was developed using Clarity Western ECL Substrate (Bio-Rad), and images were captured with the Bio-Rad ChemiDoc imaging system. β-Actin, β-Tubulin, and *GAPDH* were used as an internal loading control. Three independent experiments were performed, and protein bands were quantified using Image J, version 1.54p.

### 2.8. RNA Extraction and qPCR Analysis

Total RNA was extracted from cells using Trizol^®^ Reagent (Invitrogen, Carlsbad, CA, USA) according to the manufacturer’s instructions. Contaminating DNA in RNA samples was removed with DNase treatment using the DNA-free kit (Invitrogen). RT-qPCR was performed as a two-step process using the High-Capacity cDNA Reverse Transcription Kit and PowerTrack SYBR Green Master Mix (Applied Biosystems, Carlsbad, CA, USA). Each cDNA sample of 5ng was run in triplicate using the StepOnePlus Real Time PCR system. Primer sequences were retrieved from the Harvard PrimerBank (http://pga.mgh.harvard.edu/primerbank/ (accessed on 1 August 2025)) [33] and published studies (primers for qPCR are listed in Table A1). The Ct values were normalized with *GAPDH* as an internal control. Data is presented as a fold change from untreated cells.

### 2.9. Statistical Analysis

GraphPad Prism v.10.6.1 (GraphPad, San Diego, CA, USA) was used for statistical analysis. Two-tailed unpaired Student’s *t*-test was used for the comparison between two groups. Cox proportional hazards models were applied for Kaplan–Meier plots. One-way ANOVA or two-way ANOVA analysis followed by Dunnett’s test were applied for multiple comparisons. Values of *p* ≤ 0.05 were considered statistically significant. A power calculation was set at the 5% significance with 80% power.

## 3. Results

### 3.1. High STAT2 mRNA Expression Predicts Poor Survival in TCGA Colon Cancer Cohort

Several studies have reported that STAT2 is required for mediating the antitumor effects of type I IFNs [32,34]. However, because we previously showed that STAT2 promotes tumor development in a model of colitis-associated cancer [15] and given that IFN-I signaling was established as tumor-suppressive and a predictor of overall survival in colon cancer [35,36,37] with STAT2 inactivation [11], we sought to determine whether STAT2 exhibits a similar or distinct association with patient outcomes. Analysis of TCGA colon adenocarcinoma (COAD) data revealed that STAT2 expression was significantly elevated in tumor tissues compared with normal colon samples (Figure 1a; *p* = 0.00012), a pattern that was also observed for IFNAR1 (Figure 1c; *p* = 0.0046). Kaplan–Meier survival analysis was then performed using the same TCGA-COAD dataset to evaluate the prognostic relevance of STAT2 expression. Patients were stratified into high- and low-STAT2 expression groups based on the median expression value (Figure 1b). The analysis showed that patients with high STAT2 expression had significantly reduced overall survival over a five-year period compared with those with low expression (log-rank test *p* = 0.049). Furthermore, Kaplan–Meier analysis of STAT2-high tumors stratified by IFNAR1 expression (STAT2-high/IFNAR1-low vs. STAT2-high/IFNAR1-high) did not demonstrate a statistically significant difference in survival (log-rank *p* = 0.56) even though the groups appeared to diverge visually approximately 24 months after diagnosis (Figure 1d). These findings suggest that elevated STAT2 expression may serve as a negative prognostic indicator in colon cancer.

### 3.2. STAT2 and IFNAR1 Deletions Differentially Affect Downstream STAT Activation

To compare the functional consequences of STAT2 and IFNAR1 deletion, we first generated human HCT116 cells with individual deletions of each gene and confirmed successful knockout by western blotting and gene expression analyses (Figure 2a–c, Appendix A). Analysis of basal protein levels revealed that total expression of STAT1, STAT2, and STAT3 remained largely unaffected in both STAT2 KO and IFNAR1 KO tumor cells (Figure 2a), indicating that deletion of either gene does not alter steady-state STAT2 protein abundance. However, while both STAT2 KO and IFNAR1 KO cells lacked canonical IFN-I responses, a marked difference was observed in STAT1 and STAT3 activation. Phosphorylation of STAT1 was reduced while phosphorylation of STAT3 was enhanced in STAT2 KO cells, indicating that STAT2 deletion paradoxically preserves STAT3 signaling. As expected, IFNAR1 deletion impaired the activation of all three STATs: STAT1, STAT2, and STAT3. Functional validation by qPCR confirmed that IFN-induced transcriptional response of canonical STAT2-dependent target genes (*ISG15* and *IFIT1*) was impaired in both knockout lines (Figure 2c), demonstrating that although STAT2 and IFNAR1 deletion both disrupt IFN-I-mediated gene *GAPDH* expression, STAT2 deletion uniquely preserves STAT3 activation. These results highlight distinct roles for STAT2 and IFNAR1 in regulating downstream STAT signaling in colon cancer cells.

### 3.3. STAT2 and IFNAR1 Deletions Have Opposing Effects on Human Colon Cancer Cell Proliferation and Tumor Growth

To determine whether the differential signaling profiles of STAT2 KO and IFNAR1 KO cells were translated into distinct growth phenotypes, we next examined their proliferative capacity in vitro and in vivo. In an MTS cell proliferation assay, STAT2 KO cells displayed significantly reduced proliferation compared with parental controls, from a 4-fold to a 3-fold change, whereas IFNAR1 KO cells showed no significant difference from wild-type cells (Figure 3a). To assess the impact of STAT2 and IFNAR1 loss on tumorigenicity, we implanted each cell line subcutaneously into immunodeficient *Rag1 KO* mice and monitored tumor growth over time. Consistent with the in vitro findings, STAT2 KO cells formed markedly smaller tumors, while IFNAR1 KO cells established significantly larger tumors compared with controls (Figure 3b). Moreover, overexpression of STAT2 in colon cancer cells accelerated tumor growth in vivo (Figure 3c), further supporting a tumor-promoting role for STAT2. Together, these results demonstrate that, despite their shared involvement in IFN-I signaling, STAT2 and IFNAR1 exert opposing effects on colon cancer cell proliferation and tumor progression.

### 3.4. Distinct IFN-I Signaling Defects in STAT2- and IFNAR1-Deficient Murine Colon Carcinoma Cells

To determine whether the differential effects of STAT2 and IFNAR1 deletion observed in human cells were conserved in a murine context, we evaluated STAT2- and IFNAR1-deficient mouse MC38 colon carcinoma cell lines. Western blot analysis confirmed successful deletion of each gene, with total basal levels of STAT1, STAT2, and STAT3 remaining steady across all genotypes (Figure 4a,b, Appendix A). Upon IFN-I stimulation, we observed a distinct alteration in downstream signaling: STAT1 activation was completely absent in STAT2 KO cells (Figure 4a and Appendix A), while STAT3 phosphorylation was noticeably reduced compared with parental cells (Figure 4b and Appendix A). No activation of STAT1 and STAT3 was seen in IFNAR1 KO cells as expected (Figure 4a,b, Appendix A). The attenuated STAT3 activation in STAT2 KO cells may, in part, reflect the presence of mutant *p53* in this cell line, which has been reported to influence STAT3 signaling [38,39]. Despite these differences in STAT activation, qPCR analysis confirmed that both STAT2 KO and IFNAR1 KO cells exhibited defective induction of canonical IFN-stimulated genes following IFN-I treatment (Figure 4c). Notably, activation of STAT2 by IFN-I could not be assessed because there are no reliable commercial sources that detect the murine form of tyrosine phosphorylated STAT2. Together, these results indicate that, like the human model, loss of STAT2 or IFNAR1 disrupts IFN-I-dependent transcription, although the downstream signaling consequences differ between the two genes in murine colon carcinoma cells.

### 3.5. Reduced Proliferation and Tumorigenicity in STAT2-Deficient Murine Colon Cancer Cells

To assess the functional impact of *Stat2* and *Ifnar1* deletion on tumor cell growth, we evaluated their proliferative capacity in vitro and in vivo. In an MTS assay, STAT2 KO cells displayed significantly reduced proliferation compared with parental controls, whereas IFNAR1 KO cells showed no statistically significant difference in growth rate when measured over 72 h (Figure 5a). To determine whether these effects were maintained in vivo, each cell line was implanted subcutaneously into immunocompetent mice, and tumor growth was monitored over time. Consistent with the in vitro results, STAT2 KO cells formed markedly smaller tumors (Figure 5b), while parental and IFNAR1 KO cells developed tumors of comparable size (Figure 5c). These findings demonstrate that loss of STAT2 impairs both cell proliferation and tumor growth, whereas loss of IFNAR1 does not, further supporting the notion that STAT2 promotes tumorigenicity through mechanisms distinct from canonical IFNAR1 signaling.

## 4. Discussion

In this study, we provide experimental evidence that STAT2 plays a tumor-promoting role in colon cancer that is distinct from the canonical, tumor-suppressive effects of IFNAR1-mediated IFN-I signaling. Through integrated analyses of human TCGA-COAD datasets and complementary functional studies in human and murine colon carcinoma models, we show that the expressions of STAT2 and IFNAR1 are elevated in tumors relative to normal colon tissue, and high STAT2 expression correlates with reduced overall survival. Although stratification of tumors with high STAT2 expression by low versus high IFNAR1 levels did not reveal statistically significant differences in patient survival, previous studies have demonstrated that reduced IFNAR1 expression, specifically in cytotoxic T cells, is associated with poor prognosis in colon cancer [11,35]. Together, these observations suggest that the tumor-promoting effects of STAT2 may operate in a tumor-cell intrinsic manner, rather than being driven by differences in IFNAR1-dependent immune signaling. Notably, high STAT2 expression has been reported to be associated with reduced survival in other types of cancer [18,19], which has prompted further investigation, and recent studies support the notion that STAT2 may act as a tumor promoter [22,23,40,41]. Functionally, our study shows that loss of STAT2 suppressed tumor cell proliferation and impaired tumor growth in vivo, whereas loss of IFNAR1 produced the opposite effect, resulting in enhanced tumor growth. These findings suggest that STAT2 contributes to tumor progression through mechanisms independent of classical IFNAR1 signaling.

To further dissect the molecular basis of this divergence, we compared how loss of STAT2 vs. IFNAR1 affects downstream IFN-I signaling and STAT pathway activation. Although both STAT2 and IFNAR1 are integral components of the IFN-I signaling cascade, our results reveal fundamental differences in how their loss alters downstream responses. Deletion of either gene disrupted IFN-I-induced transcriptional responses, confirming impaired canonical IFN-I signaling. However, STAT2 deletion uniquely preserved STAT3 activation. The retention of STAT3 signaling in STAT2 KO cells suggests that STAT2 may normally restrain or modulate alternative STAT pathways under basal or stress conditions. Indeed, STAT2 has been reported to inhibit STAT1-dependent signaling pathways [42]. Given that STAT3 activation is frequently associated with oncogenic signaling and suppression of antitumor immunity [43,44], the persistence of IFN-I-induced STAT3 phosphorylation in STAT2 KO cells may represent a compensatory or context-specific adaptation. For example, activation of STAT3 by IFN-I has been shown to enhance cytolytic T cell function [35].

Interestingly, these signaling distinctions were observed in both human and murine colon cancer cells, despite some model-specific variations. In murine STAT2 KO cancer cells, reduced STAT3 activation was accompanied by loss of STAT1 phosphorylation, which may reflect the influence of mutant p53 on STAT3 regulation in this cell line [38,39] in contrast to HCT116, which carry wild-type p53. Nonetheless, the overarching pattern remained consistent: loss of STAT2 suppressed cell proliferation and tumor growth, whereas loss of IFNAR1 did not, underscoring their divergent roles in tumor biology.

Our findings challenge the conventional view of STAT2 as solely a mediator of IFN-I-dependent antitumor immunity and instead support a context-dependent function for STAT2 as a tumor promoter in colon cancer. The dual nature of STAT2 may depend on its interactions with other transcription factors, such as STAT3, IRF family members, and NFκB [45,46,47], or interactions with circular RNA CAPRIN1, which promotes tumor progression and lipid synthesis [41], as well as the inflammatory milieu within the tumor microenvironment. Although the exact mechanism by which STAT2 promotes colorectal cancer remains to be determined, there is some mechanistic evidence that STAT2 can act outside the canonical IFNAR1-STAT2/STAT1/IRF9 axis via unphosphorylated STAT2. One potential mechanistic explanation involves STAT3 as a compensatory pathway, where STAT2/IRF9 associates with the p65 subunit of NFκB, induces IL-6 expression, and ultimately activates STAT3 [15,47]. Future experiments, such as performing a ChIP assay for STAT2 to identify target genes with tumorigenic function and using mutants of STAT2 that cannot be tyrosine phosphorylated, will provide initial mechanistic insight into how STAT2 may promote tumor growth.

A recent report indicated that chemoresistance in ovarian cancer is mediated through a FBN1/VEGF/STAT2 signaling axis [21]. Other studies have shown that STAT2 promotes chemoresistance [16,48] driven by increased STAT2 expression [49], an important observation that warrants further investigation, given that high STAT2 levels in colon cancer correlate with reduced overall survival. Moreover, the tumor-promoting role of STAT2 identified here aligns with emerging evidence in other diseases, where STAT2 has been implicated in sustaining chronic inflammation [15,50,51,52]. It is noteworthy to mention that individuals born with a STAT2 deficiency have susceptibility to viral infections in early childhood, and those who live into adulthood have reduced severity of infections over time [53]. Although it is too early to predict, it would be interesting to know if later in life these individuals have a lower risk of developing colorectal cancer. In summary, our study reveals that STAT2 promotes colon tumorigenesis independently of canonical IFNAR1-mediated signaling.

## 5. Conclusions

Our findings demonstrate that STAT2 functions as a tumor-promoting factor in colon cancer, acting independently of canonical IFNAR1-mediated type I interferon signaling. While IFNAR1 loss did not accelerate tumor growth, STAT2 deletion consistently suppressed cell proliferation and tumor formation in both human and murine models. These results reveal a context-dependent, pro-tumorigenic role for STAT2 and underscore the complexity of the IFN-I signaling axis in colorectal cancer. Targeting STAT2 or its downstream pathways may, therefore, represent a novel therapeutic strategy for mitigating colon cancer progression.

## Figures and Tables

**Figure 1 curroncol-32-00707-f001:**
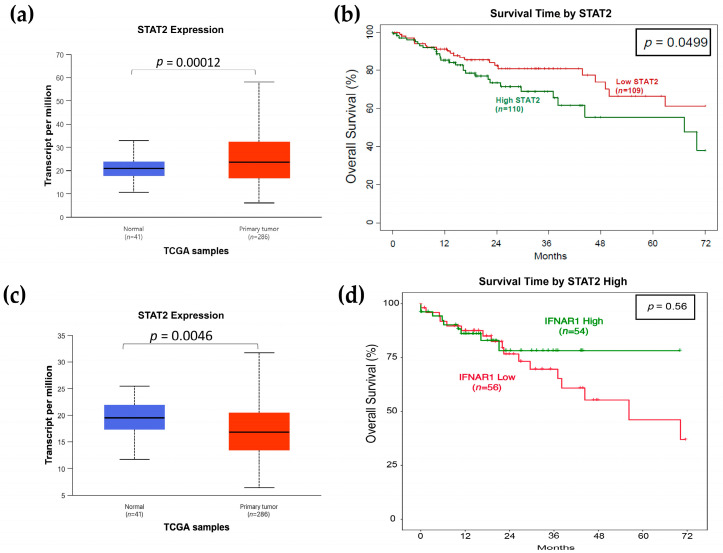
STAT2 expression is elevated in colon tumors and correlates with reduced survival. Boxplots showing (**a**) STAT2 and (**c**) IFNAR1 mRNA expression are significantly higher in TCGA-COAD tumor tissues (*n* = 286) compared with normal colon samples (*n* = 41). (**b**) Kaplan–Meier survival curves for patients stratified by median STAT2 expression show reduced overall survival in the STAT2-high group. (**d**) STAT2-high tumors stratified by IFNAR1-high (*n* = 56) and IFNAR-low (*n* = 54) expression show no survival difference. Cox proportional hazards model was applied. Statistical significance; *p* ≤ 0.05.

**Figure 2 curroncol-32-00707-f002:**
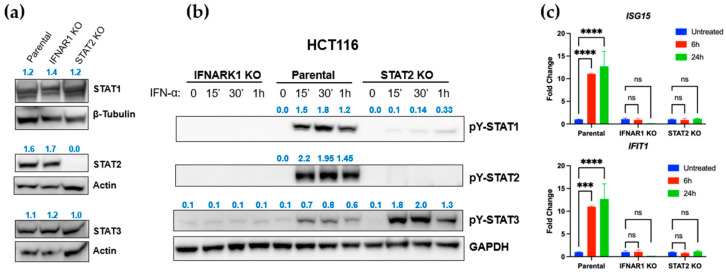
STAT3 activation is preserved in STAT2 KO tumor cells. (**a**) Western blots confirming basal levels of STAT1, STAT2, and STAT3 after deletion of STAT2 or IFNAR1 in knockout clones. (**b**) Time course analyses of phosphorylated STATs following IFN-I stimulation. (**c**) Impaired transcriptional response to IFN-I in both STAT2 KO and IFNAR1 KO cells. *** *p* ≤ 0.001; **** *p* ≤ 0.0001. ns, not statistically significant.

**Figure 3 curroncol-32-00707-f003:**
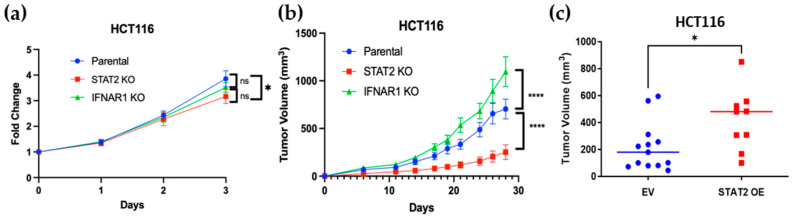
STAT2 and IFNAR1 differentially regulate colon cancer cell proliferation and tumor growth. (**a**) In vitro MTS showing reduced proliferation of STAT2 KO compared with parental and IFNAR1 KO HCT116 cells over the course of 72 h. Data represent mean ± SEM from *n* = 3. (**b**) Growth of tumor xenografts in immunodeficient *Rag1KO* mice injected subcutaneously with parental, STAT2 KO, or IFNAR1 KO cells. (**c**) Overexpression of STAT2 in HCT116 cells enhanced tumor growth in vivo (*n* = 5–8 mice per study). * *p* ≤ 0.05; **** *p* ≤ 0.0001. ns, not statistically significant. Results are from two combined independent experiments.

**Figure 4 curroncol-32-00707-f004:**
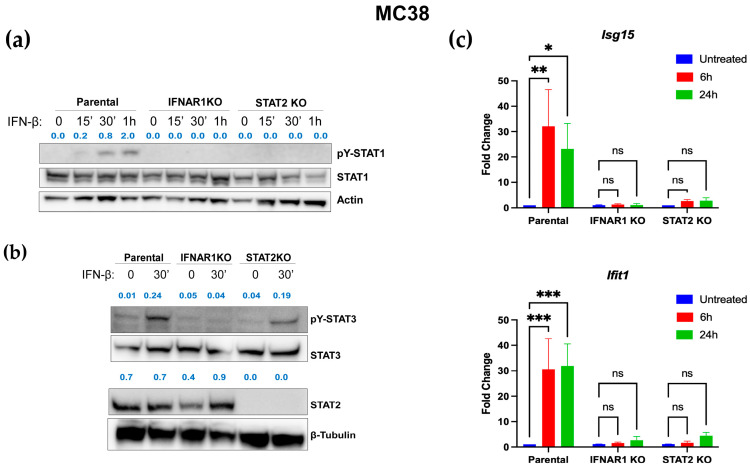
STAT2 signaling and IFN-I responsiveness in STAT2- and IFNAR1-deficient murine colon carcinoma cells. (**a**) Western blot analyses show IFN-I-stimulated phosphorylation of STAT1 and basal STAT1 expression in parental, STAT2 KO, and IFNAR1 KO MC38 cells. (**b**) IFN-I-stimulated phosphorylation of STAT3 and basal STAT2 expression analyzed by western blot analysis. (**c**) Impaired transcriptional responses in both KO cell lines after 6 and 24 h of IFN-β treatment. Data are shown as fold change from corresponding untreated cell genotype. * *p* ≤ 0.05; ** *p* ≤ 0.01; *** *p* ≤ 0.001. ns, not statistically significant.

**Figure 5 curroncol-32-00707-f005:**
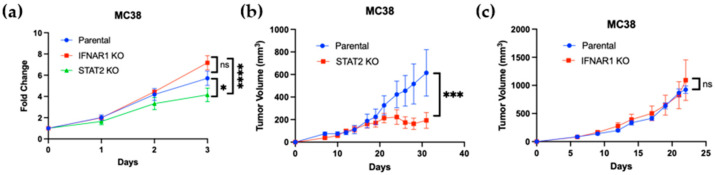
Loss of STAT2, but not IFNAR1, reduces proliferation and tumor growth in murine colon carcinoma cells. (**a**) In vitro MTS assay showing reduced proliferation of Stat2 KO tumor cells compared with parental and IFNAR1 KO cell lines over 72 h. Data represent mean ± SEM from *n* = 3. (**b**,**c**) Tumor growth curves of wild-type mice injected subcutaneously with parental, STAT2 KO, or IFNAR1 KO tumor cells. * *p* < 0.05; *** *p* < 0.001; **** *p* < 0.0001. Data are shown as mean ± SEM from *n* = 6–8 mice per group.

## Data Availability

The data presented in this study are available on request due to privacy concerns.

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
