# Peer review of "STAT2 Promotes Tumor Growth in Colorectal Cancer Independent of Type I IFN Receptor Signaling"

_curroncol, 2025, doi:10.3390/curroncol32120707_

Round 1

Reviewer 1 Report

Comments and Suggestions for Authors

This manuscript addresses an important question: whether STAT2 promotes colorectal cancer progression independently of canonical type I interferon (IFN-I) signaling via IFNAR1. The authors combine TCGA bioinformatics, CRISPR gene editing, in vitro assays, and in vivo xenografts in both human and murine models to reach the conclusion that STAT2 has a cell-intrinsic tumor-promoting function distinct from IFNAR1 signaling. However, several major issues limit the clarity, rigor, and impact of the study.

  1. Mechanistic insight is insufficient and remains speculative. The central conclusion—that STAT2 promotes tumor growth independently of IFNAR1—is supported by phenotypes, but mechanistic evidence is lacking. The manuscript frequently cites STAT3 or alternative pathway compensation, but no rescue or inhibition experiments were performed.

  1. The manuscript frequently uses p-values but lacks: statistical tests for every figure, clarifications on replicates (biological vs technical), confidence intervals or effect sizes.

  1. The authors state that IFNAR1 loss correlates with poor prognosis in CRC, yet no direct comparison is made between STAT2-high / IFNAR1-low subgroups in TCGA.

  1. Missing catalog numbers for several reagents (e.g., IFN). Include actual values (e.g., fold changes, p-values) instead of only stating “significant”.

Reviewer 2 Report

Comments and Suggestions for Authors

The aim of this original paper was to „investigate the relationship between Signal Transducer and Activator of Transcription 2 (STAT2) expression and patient survival in colorectal cancer (CRC) and examined the functional role of STAT2 in tumor growth using preclinical tumor models”. In addition, the authors wanted to evaluate whether STAT2’s tumor-promoting activity depends on canonical type I interferon (IFN-I) signaling.

I have read the paper carefully and I have only a few comments.

The work is interesting and shows the dual role of STAT2 in CRC carcinogenesis. The introduction is clearly written, and the thesis of the work are presented in an understanding form.

The materials and methods of the work are also presented in a professional manner. Only in line 156, please correct the name of the gene “Gapdh” to capital letters, as in other parts of the work.

The results are presented in the form of graphs, figures, tables, etc., generally in a clear manner, although there are minor inaccuracies.

Figure 1a itself gives a value of P=1.16 x 104, which is not clear. And in the description of Figure 1 (at the bottom), p<0.05 and p<0.0001 are given, but it is not clear what these significance levels refer to. Please complete or correct this.  In Figure 1b, the value p=0.0499, but this is p=0.05 and not below 0.05.

In the description of Figure 2 (line 210), please correct the number of asterisks and their explanations. ** - p<0.01; *** - p<0.001 etc.

In line 231, please correct the name of the cell line “HCTT116” to HCT116.

Figure 4 - the letter (c) is missing from the figure itself, but there is a description for this letter below; please check.

The discussion is adequate for the results obtained. The authors refer to the results of other authors' research and their own work from previous years (No. 15). The discussion is specific and the publications cited are correct.

I think that the only thing that could be checked is the language of the discussion itself, e.g., avoiding re-explaining abbreviations that have already been explained, such as IFN-I (line 287), removing unnecessary spaces (lines: 21, 294, 297) or adding them (line 328), etc.

The conclusions of the work are correct, no comments.

References - please correct the whole thing according to the publisher's recommendations, as there is complete chaos here.

Round 2

Reviewer 1 Report

Comments and Suggestions for Authors

I don't have further comments at current stage.